# A Finite Element Model for Dynamic Analysis of Triple-Layer Composite Plates with Layers Connected by Shear Connectors Subjected to Moving Load

**DOI:** 10.3390/ma12040598

**Published:** 2019-02-16

**Authors:** Hoang-Nam Nguyen, Tan-Y. Nguyen, Ke Van Tran, Thanh Trung Tran, Truong-Thinh Nguyen, Van-Duc Phan, Thom Van Do

**Affiliations:** 1Modeling Evolutionary Algorithms Simulation and Artificial Intelligence, Faculty of Electrical & Electronics Engineering, Ton Duc Thang University, Ho Chi Minh 700000, Vietnam; nguyenhoangnam@tdtu.edu.vn; 2Faculty of Mechanical Engineering, University of Technical Education, Ho Chi Minh 700000, Vietnam; ynt.ncs@hcmute.edu.vn (T.-Y.N.); thinhnt@hcmute.edu.vn (T.-T.N.); 3Faculty of Information Technology, Malaysia University of Science and Technology (MUST), Petaling Jaya, Selangor 47810, Malaysia; 4Faculty of Mechanical Engineering, Le Quy Don Technical University, Ha Noi 100000, Vietnam; tranke92@gmail.com (K.V.T.); tranthanh0212@gmail.com (T.T.T.); 5Center of Excellence for Automation and Precision Mechanical Engineering, Nguyen Tat Thanh University, Ho Chi Minh 700000, Vietnam

**Keywords:** triple-layer composite plate, Mindlin’s theory, finite element model, moving load

## Abstract

Triple-layered composite plates are created by joining three composite layers using shear connectors. These layers, which are assumed to be always in contact and able to move relatively to each other during deformation, could be the same or different in geometric dimensions and material. They are applied in various engineering fields such as ship-building, aircraft wing manufacturing, etc. However, there are only a few publications regarding the calculation of this kind of plate. This paper proposes novel equations, which utilize Mindlin’s theory and finite element modelling to simulate the forced vibration of triple-layered composite plates with layers connected by shear connectors subjected to a moving load. Moreover, a Matlab computation program is introduced to verify the reliability of the proposed equations, as well as the influence of some parameters, such as boundary conditions, the rigidity of the shear connector, thickness-to-length ratio, and the moving load velocity on the dynamic response of the composite plate.

## 1. Introduction

In recent years, composite layered systems have attracted a great of interest from many researchers due to their optimized material configuration, such as their high strength-to-weight ratio and stiffness-to-weight ratio. For a multi-layer composite beam, all of its layers are connected by shear connectors, which play a very crucial role in the mechanical behavior of the beam. Newmark et al. [1] proposed the governing differential equations for elastically connected steel-concrete beams, based on the linear elastic Euler–Bernoulli beam theory. Various later works [2,3,4] improved the shear effect of Newmark’s model by using Timoshenko composite beam theory. Silva et al. [5] and Nguyen et al. [6,7] utilize the finite element method (FEM) to analyze the linear static properties of multi-layered composite beams. Later on, Newmark’s model was developed for dynamic and non-linear problems [8,9,10,11,12].

In addition, for the Timoshenko beam theory (TBT), Chakrabarti et al. [13,14] introduced static analysis of two-layer composite beams by using higher order beam theory (HBT). The dynamic response of composite beams with shear connectors was computed using the finite element method (FEM) and higher-order beam theory [15]. HBT has partially overcome the side effect of shear correction factor. Additionally, Subramanian [16] used HBT and FEM for dynamic analysis of laminated composite beams. Li et al. [17] studied the free vibration of axially loaded composite beams with general boundary conditions using hyperbolic shear deformation theory. Vo and Thai [18] presented the static behavior of composite beams using various refined shear deformation theories [19].

Most of the higher order theories, including Reddy’s higher order beam theory (RHBT) [20], tend to ignore the transverse deformation of multi-layer beams. Manjunatha and Kant [21] proposed to use C^0^ finite element [22,23] and a set of higher order theories for the analysis of composite and sandwich beams. Kant’s theories incorporate the non-linear variation of displacement through beam thickness to eliminate the use of shear correction coefficients. Yan et al. [24] developed a three-dimensional damage plasticity based on the finite element model (FEM) to simulate the ultimate strength behavior of the SCS sandwich structure under concentrated loads.

Carrena [25] worked on the assumption that the displacement field is expanded in terms of generic functions, which is the unified formulation by Carrera (CUF) [26], to examine the static response of beams with different cross sections, such as square, C-shaped and bridge-like sections. Based on the mentioned approach, Cinefra et al. [27] employed MITC9 (Mixed Interpolated of Tensorial Components using 9 nodes) shell elements to analyze the mechanical behavior of laminated composite plates and shells. Muresan et al. [28] carried out research on the stability of the thin walled prismatic bars based in the generalized beam theory (GBT), which is an efficient approach developed by Schardt [29]. Yu et al. [30] used the variational asymptotic beam section analysis (VABS) for mechanical behavior of various cross sections, such as the elliptic and triangular sections.

The dynamic behaviors of plates under moving loads are also interesting problems in engineering such as bridges and roads, space vehicles, submarines and mechanical engineering and so on. So many scholars have studied on this aspect in past decades. Ouyang [31] briefly reviewed a variety of moving-load problems and several analytical solution methods. Fryba [32] has summarized a variety of engineering problems that analyzed the dynamics of structures under moving loads. Song et al. [33] proposed a novel method to predict the dynamic behaviors of flat plate of arbitrary boundary conditions subjected to moving loads, based on the Kirchhoff plate theory.

Although there are plenty of published works dealing with the non-linear, linear, static and dynamic problems of composite and sandwich beams using shear connectors, to our knowledge, there seems to be no analysis regarding the calculation of triple-layer composite plate with layers connected by shear connectors subjected to moving load. In this paper, we combined the published theories on multi-layered beams, Mindlin’s plate theory, and finite element modelling to simulate the oscillation of triple-layer plates with layers connected by shear connectors subjected to moving loads. We also built the mechanical properties and established equations describing the dynamic response of the plates. Additionally, we studied the influence of geometric parameters, material, load, etc., on the dynamic response of the referred plate under the influence of a moving load.

## 2. Triple-Layer Composite Plates with Shear Connectors

Consider the triple-layer plate under moving load as follows in Figure 1:

The plate includes three layers: top layer (*t*), bottom layer (*b*) and intermediate layer (*c*), which can be made of the same or different materials. These layers can slide relative to each other but are not allowed to disconnect during the deformation. Each layer is attached with the local coordinate *oxyt*, *oxyc* and *oxyb.* The plate is divided into six components *h*_1_, *h*_2_, *h*_3_, *h*_4_, *h*_5_, *h*_6_, as in Figure 1. u*_t_*_0_, u*_c_*_0_ and u*_b_*_0_ are the displacements along the *x* axis; *v_t_*_0_, *v_c_*_0_ and *v_b_*_0_ represent the displacements along the *y* axis at the mid-plane of each layer. *u_tc_*, *u_cb_* are the relative displacements between (*t*) layer and (*c*) layer along *x* axis; *v_tc_*, *v_cb_* are the relative displacements between (*t*) layer and (*c*) layer along *y* axis

## 3. Finite Element Model for Dynamic Analysis of Triple-Layer Composite Plates with Shear Connectors

### 3.1. Dynamic Equation for Plate Element

According to Mindlin’s plate theory, the displacements of the *u*, *v*, *w* of each layer are shown as:(1)$$\left\{ \begin{array}{l} u_{k}=u_{k0}(x,y)+z_{k}\varphi_{k}(x,y)\\ v_{k}=v_{k0}(x,y)+z_{k}\psi_{k}(x,y)\\ w_{k}=w(x,y) \end{array}\right.\;(k=t,\;c,\;b)$$

Relative displacements between layers in contact surfaces are:

+Relative displacement between *t* and *c* layers
(2)$$\left\{ \begin{array}{l} u_{tc}(x,y)=u_{t}(x,y,h_{2})-u_{c}(x,y,-h_{3})\\ v_{tc}(x,y)=v_{t}(x,y,h_{2})-v_{c}(x,y,-h_{3}) \end{array}\right.$$

+Relative displacement between *c* and *b* layers
(3)$$\left\{ \begin{array}{l} u_{cb}(x,y)=u_{c}(x,y,h_{4})-u_{b}(x,y,-h_{5})\\ v_{cb}(x,y)=v_{c}(x,y,h_{4})-v_{b}(x,y,-h_{5}) \end{array}\right.$$

At the tangential plane between layers, we have:(4)$$\left\{ \begin{array}{l} z_{t}=h_{2};z_{c}=-h_{3}\\ z_{c}=h_{4};z_{b}=-h_{5} \end{array}\right.$$
where $h_{4}=h_{3}=\frac{h_{c}}{2}$.

Then, we obtain:(5)$$\{\begin{array}{c} u_{tc}(x,y)=u_{t0}(x,y)-u_{c0}(x,y)+h_{2}\varphi_{t}(x,y)+h_{3}\varphi_{c}(x,y)\\ v_{tc}(x,y)=v_{t0}(x,y)-v_{c0}(x,y)+h_{2}\psi_{t}(x,y)+h_{3}\psi_{c}(x,y) \end{array}$$
(6)$$\{\begin{array}{c} u_{cb}(x,y)=u_{c0}(x,y)-u_{b0}(x,y)+h_{4}\varphi_{c}(x,y)+h_{5}\varphi_{b}(x,y)\\ v_{cb}(x,y)=v_{c0}(x,y)-v_{b0}(x,y)+h_{4}\psi_{c}(x,y)+h_{5}\psi_{b}(x,y) \end{array}$$

Relation between the strains and displacements of the layers is shown as:

+For the *k^th^* layer:(7)$$\begin{array}{l} \varepsilon_{kx}=\frac{\partial u_{k}}{\partial x}=\frac{\partial u_{k0}}{\partial x}+z_{k}\frac{\partial \varphi_{k}}{\partial x};\;\\ \varepsilon_{ky}=\frac{\partial v_{k}}{\partial y}=\frac{\partial v_{k0}}{\partial y}+z_{k}\frac{\partial \psi_{k}}{\partial y};\\ \gamma_{kxy}=\frac{\partial v_{k}}{\partial x}+\frac{\partial u_{k}}{\partial y}=\frac{\partial u_{k0}}{\partial y}+\frac{\partial v_{k0}}{\partial x}+z_{k}\left(\frac{\partial \varphi_{k}}{\partial y}+\frac{\partial \psi_{k}}{\partial x}\right); \end{array}\;\begin{array}{l} \gamma_{kxz}=\frac{\partial w_{0}}{\partial x}+\frac{\partial u_{k}}{\partial z_{k}}=\frac{\partial w_{0}}{\partial x}+\varphi_{k};\\ \gamma_{kyz}=\frac{\partial w_{0}}{\partial y}+\frac{\partial v_{k}}{\partial z_{k}}=\frac{\partial w_{0}}{\partial y}+\psi_{k}; \end{array}$$

Equation (7) is rewritten in the matrix form:(8)$$\varepsilon_{k}=\left\{\begin{array}{c} \varepsilon_{kx}\\ \varepsilon_{ky}\\ \gamma_{kxy} \end{array}\right\}=\varepsilon_{k}^{0}+z_{k}\kappa_{k};\;\gamma_{k}=\left\{\begin{array}{c} \gamma_{kyz}\\ \gamma_{kzx} \end{array}\right\}$$
where:(9)$$\varepsilon_{k}^{0}=\left\{\begin{array}{c} \varepsilon_{kx}^{0}\\ \varepsilon_{ky}^{0}\\ \gamma_{kxy}^{0} \end{array}\right\}=\left\{\begin{array}{c} \frac{\partial u_{k0}}{\partial x}\\ \frac{\partial v_{k0}}{\partial y}\\ \left(\frac{\partial u_{k0}}{\partial y}+\frac{\partial v_{k0}}{\partial x}\right) \end{array}\right\};\;\kappa_{k}=\left\{\begin{array}{l} \kappa_{kx}\\ \kappa_{ky}\\ \kappa_{kxy} \end{array}\right\}=\left\{\begin{array}{l} \frac{\partial \varphi_{k}}{\partial x}\\ \frac{\partial \psi_{k}}{\partial y}\\ \frac{\partial \varphi_{k}}{\partial y}+\frac{\partial \psi_{k}}{\partial x} \end{array}\right\};\gamma_{k}=\left\{\begin{array}{l} \gamma_{kxz}\\ \gamma_{kyz} \end{array}\right\}=\left\{\begin{array}{l} \frac{\partial w_{0}}{\partial x}+\varphi_{k}\\ \frac{\partial w_{0}}{\partial y}+\psi_{k} \end{array}\right\}$$

Relation between the stress and strain of the *k^th^* layer is:(10)$$\sigma_{k}=D_{k}\varepsilon_{k};\;\tau_{k}=\frac{5}{6}G_{k}\gamma_{k}$$
where: *ν_k_* is the Poisson’s ratio of the *k^th^* layer and
(11)$$D_{k}=\frac{E_{k}}{1-v^{2}}\left[\begin{array}{ccc} 1 & \nu_{k} & 0\\ \nu_{k} & 1 & 0\\ 0 & 0 & (1-\nu_{k})/2 \end{array}\right];\;G_{k}=\frac{E_{k}}{2(1+\nu_{k})}\left[\begin{array}{cc} 1 & 0\\ 0 & 1 \end{array}\right]$$
where *E_k_* is the elasticity modulus of the *k^th^* layer.

We used an 8-node isoparametric element with 13 degrees of freedom (DOF) for each node. The DOFs of the *i*^th^ node $\left\{q_{e}^{i}\right\}$ and the plate element $\left\{q_{e}\right\}$ are defined as:(12)$$q_{e}^{i}={\left\{\begin{array}{ccccccccc} u_{t0i} & v_{t0i} & \varphi_{ti} & \psi_{ti} & u_{c0i} & v_{c0i} & \varphi_{ci} & \psi_{ci} & \;u_{b0i}\;v_{b0i}\;\varphi_{bi}\;\psi_{bi} \end{array}w\right\}}^{T};\;i=1÷8$$
(13)$$q_{e}={\left\{\begin{array}{cccc} q_{e}^{1} & q_{e}^{2} & q_{e}^{3} & q_{e}^{4}\;\begin{array}{cccc} q_{e}^{5} & q_{e}^{6} & q_{e}^{7} & q_{e}^{8} \end{array} \end{array}\right\}}^{T}$$
(14)$$\begin{array}{l} u_{k0}=\sum_{i=1}^{8}N_{i}\left(\xi ,\eta \right)u_{k0i};\;v_{k0}=\sum_{i=1}^{8}N_{i}\left(\xi ,\eta \right)v_{k0i}\\ \varphi_{k}=\sum_{i=1}^{8}N_{i}\left(\xi ,\eta \right)\varphi_{ki};\;\psi_{k}=\sum_{i=1}^{8}N_{i}\left(\xi ,\eta \right)\psi_{ki};\;w=\sum_{i=1}^{8}N_{i}\left(\zeta ,\eta \right)w_{i} \end{array}\;(k=t,\;c,\;b)$$
where *N_i_* (i=1÷8) is specified in Appendix A.

Then, we have the strain of elements in the *k*^th^ layer as:(15)$$\left\{ \begin{array}{l} \varepsilon_{k}=\left(B_{k}^{0}+z_{k}B_{k}^{1}\right)q_{e}\\ \gamma_{k}=S_{k}q_{e} \end{array}\right.\;(k\;=\;t,\;c,\;b)$$
where $B_{ki}^{0}$, $B_{ki}^{1}$, $S_{k}$ are defined as:(16)$$\begin{array}{l} B_{k}^{0}=\left[\begin{array}{cccc} B_{k1}^{0} & B_{k2}^{0} & B_{k3}^{0} & B_{k4}^{0}\;\begin{array}{cccc} B_{k5}^{0} & B_{k6}^{0} & B_{k7}^{0} & B_{k8}^{0} \end{array} \end{array}\right];\\ B_{k}^{1}=\left[\begin{array}{cccc} B_{k1}^{1} & B_{k2}^{1} & B_{k3}^{1} & B_{k4}^{1}\;\begin{array}{cccc} B_{k5}^{1} & B_{k6}^{1} & B_{k7}^{1} & B_{k8}^{1} \end{array} \end{array}\right];\\ S_{k}=\left[\begin{array}{cccc} S_{k1} & S_{k2} & S_{k3} & S_{k4}\;\begin{array}{cccc} S_{k5} & S_{k6} & S_{k7} & S_{k8} \end{array} \end{array}\right]; \end{array}$$

Note that $B_{ki}^{0}$, $B_{ki}^{1}$, $S_{ki}$ are shown in Appendix B

The elastic force of shear connectors per length unit is:

+For *t* and *c* layers
(17)$$F_{e}^{tc}={\left\{\begin{array}{l} F_{eu}\\ F_{ev} \end{array}\right\}}_{ct}=k_{tc}\left[\begin{array}{cc} 1 & 0\\ 0 & 1 \end{array}\right]\left\{\begin{array}{l} u_{tc}\\ v_{tc} \end{array}\right\}=K_{e}^{tc}q_{e}^{tc}$$
where:(18)$$q_{e}^{tc}=\left\{\begin{array}{c} u_{tc}\\ v_{tc} \end{array}\right\}=\left[\begin{array}{c} u_{t0}+h_{2}\varphi_{t}-u_{c0}+h_{3}\varphi_{c}\\ v_{t0}+h_{2}\psi_{t}-v_{c0}+h_{3}\psi_{c} \end{array}\right]=N_{tc}q_{e}=\sum_{i=1}^{8}{\left(N_{tc}\right)}_{i}q_{e}^{i}$$
in which:(19)$${\left(N_{tc}\right)}_{i}=\left[\begin{array}{ccccccccc} N_{i} & 0 & h_{2}N_{i} & 0 & -N_{i} & 0 & h_{3}N_{i} & 0 & 0\;0\;0\;0\;0\\ 0 & N_{i} & 0 & h_{2}N_{i} & 0 & -N_{i} & 0 & h_{3}N_{i} & 0\;0\;0\;0\;0 \end{array}\right]$$

+For *b* and *c* layers
(20)$$F_{e}^{cb}={\left\{\begin{array}{l} F_{eu}\\ F_{ev} \end{array}\right\}}_{cb}=k_{cb}\left[\begin{array}{cc} 1 & 0\\ 0 & 1 \end{array}\right]\left\{\begin{array}{l} u_{cb}\\ v_{cb} \end{array}\right\}=K_{e}^{cb}q_{e}^{cb}$$
where:(21)$$q_{e}^{cb}=\left\{\begin{array}{c} u_{cb}\\ v_{cb} \end{array}\right\}=\left[\begin{array}{c} u_{c0}+h_{4}\varphi_{c}-u_{b0}+h_{5}\varphi_{b}\\ v_{c0}+h_{4}\psi_{c}-v_{b0}+h_{5}\psi_{b} \end{array}\right]=N_{cb}q_{e}=\sum_{i=1}^{8}{\left(N_{cb}\right)}_{i}q_{e}^{i}$$
in which:(22)$${\left(N_{cb}\right)}_{i}=\left[\begin{array}{ccccccccccccc} 0 & 0 & 0 & 0 & N_{i} & 0 & h_{4}N_{i} & 0 & -N_{i} & 0 & h_{5}N_{i} & 0 & 0\\ 0 & 0 & 0 & 0 & 0 & N_{i} & 0 & h_{4}N_{i} & 0 & -N_{i} & 0 & h_{5}N_{i} & 0 \end{array}\right]$$

In the above equations, *k_tc_*; *k_cb_* are the shear rigidity coefficients of shear connectors per length unit.

We applied the principle of virtual work to the forces applied to the plate elements:(23)$$\begin{array}{l} \sum_{k=t,c,b}\int_{V_{k}}\delta {\dot{u}}_{k}^{T}\rho_{k}{\dot{u}}_{k}dV_{k}+\sum_{k=t,c,b}\int_{V_{k}}\delta \varepsilon_{k}^{T}\sigma_{k}dV_{k}+\frac{5}{6}\sum_{k=t,c,b}\int_{V_{k}}\delta \gamma_{k}^{T}\tau_{k}dV_{k}+\sum_{k=tc,cb}\int_{A_{k}}\delta {\left(q_{e}^{k}\right)}_{T}F_{e}^{k}dA_{k}\\ -\delta q_{e}^{T}\int_{A_{t}}N_{w}p(t)dA_{t}=0 \end{array}$$

By substituting Equations (1), (15), (17) and (20) into Equation (23), we obtained the dynamic equation of the plate element as follows:(24)$$M_{e}{\ddot{q}}_{e}+K_{e}q_{e}=F_{e}(t)$$

With
(25)$$\begin{array}{l} K_{e(104x104)}=\sum_{k=t,c,b}\int_{A_{k}}{\left(B_{k}^{0}\right)}^{T}D_{k0}B_{k}^{0}dA_{k}+\sum_{k=t,c,b}\int_{A_{k}}{\left(B_{k}^{0}\right)}^{T}D_{k1}B_{k}^{1}dA_{k}+\\ +\sum_{k=t,c,b}\int_{A_{k}}{\left(B_{k}^{1}\right)}^{T}D_{k1}B_{k}^{0}dA_{k}+\sum_{k=t,c,b}\int_{A_{k}}{\left(B_{k}^{1}\right)}^{T}D_{k2}B_{k}^{1}dA_{k}+\frac{5}{6}\sum_{k=t,c,b}\int_{A_{k}}S_{k}^{T}G_{k}S_{k}dA_{k}\\ +\int_{A_{tc}}N_{tc}^{T}K_{tc}^{e}N_{tc}dA_{tc}+\int_{A_{cb}}N_{cb}^{T}K_{cb}^{e}N_{cb}dA_{cb} \end{array}$$
in which
(26)$$\left(D_{k0};\;D_{k1};\;D_{k2}\right)=\int_{-h_{k}/2}^{h_{k}/2}\left(1;\;z_{k};\;z_{k}^{2}\right)D_{k}\;dz_{k};\;H_{k}=\int_{-h_{k}/2}^{h_{k}/2}G_{k}\;dz_{k}\;(k\;=t,\;c,\;b)$$
(27)$$M_{e(104x104)}=\sum_{k=t,c,b}\int_{A_{k}}\int_{-h_{k}/2}^{h_{k}/2}L_{k}^{T}\rho_{k}L_{k}dz_{k}dA_{k}$$
where $L_{k}$ can be seen in Appendix C
(28)$$F_{e}{\left(t\right)}_{\left(104x1\right)}=\int_{A_{t}}p(t)N_{w}^{T}dA_{t}$$
in which
(29)$$N_{w}=\left[\begin{array}{cccc} N_{w1} & N_{w2} & N_{w3} & N_{w4}\;\begin{array}{cccc} N_{w5} & N_{w6} & N_{w7} & N_{w8} \end{array} \end{array}\right]$$

With
(30)$$N_{w\;j}=\left[\begin{array}{ccccccccc} 0 & 0 & 0 & 0 & 0 & 0 & 0 & 0\;0\;0\;0\;0 & N_{j} \end{array}\right]$$

In the case of taking into account structural damping, we have the force vibration equation of the plate element as follows:(31)$$M_{e}{\ddot{q}}_{e}+C_{e}{\dot{q}}_{e}+K_{e}q_{e}=F_{e}(t)$$
in which $C_{e}=\alpha M_{e}+\beta K_{e}$ and *α*, *β* are the Rayleigh drag coefficients defined in [32].

### 3.2. Formulation of the Nodal Element Load Vector

In general cases, one considers the moving load as: particle *m* moves on a plate element with a non-constant velocity *v* in a known trajectory; the load *Q* is applied on the moving particle in the direction perpendicular to the plane of the plate element, as shown in Figure 1.

Let *w*(*x*,*y*,*t*) be the bending deflection of the plate under the moving load with mass *m* (kg). The force applied on the moving load at position (*x* = *ξ*; *y* = *η*) is [32]:(32)$$R(x,y,t)=Q(t)-m\frac{d^{2}w(x,y,t)}{dt^{2}}|\begin{array}{l} x=\xi ;y=\eta \end{array}$$
where: $\frac{d^{2}w(x,y,t)}{dt^{2}}$, which is the absolute acceleration in the *z* direction at the position suffering moving load, is shown via the displacement vector, velocity and acceleration of the node as:(33)$$\begin{array}{l} \frac{d^{2}w(x,y,t)}{dt^{2}}=N^{ml}{\ddot{q}}_{e}+2\left(\dot{x}\frac{\partial N^{ml}}{\partial x}+\dot{y}\frac{\partial N^{ml}}{\partial y}\right){\dot{q}}_{e}+\ldots \\ \left({\dot{x}}^{2}\frac{\partial^{2}N^{ml}}{\partial x^{2}}+{\dot{y}}^{2}\frac{\partial^{2}N^{ml}}{\partial y^{2}}+2\dot{x}\dot{y}\frac{\partial^{2}N^{ml}}{\partial x\partial y}+\ddot{x}\frac{\partial N^{ml}}{\partial x}+\ddot{y}\frac{\partial N^{ml}}{\partial y}\right)q_{e} \end{array}$$

Replace Equation (33) into Equation (32), we have:(34)$$R(x,y,t)=Q(t)-m\left(\begin{array}{l} N^{ml}{\ddot{q}}_{e}+2\left(\dot{x}\frac{\partial N^{ml}}{\partial x}+\dot{y}\frac{\partial N^{ml}}{\partial y}\right){\dot{q}}_{e}+\ldots \\ \left({\dot{x}}^{2}\frac{\partial^{2}N^{ml}}{\partial x^{2}}+{\dot{y}}^{2}\frac{\partial^{2}N^{ml}}{\partial y^{2}}+2\dot{x}\dot{y}\frac{\partial^{2}N^{ml}}{\partial x\partial y}+\ddot{x}\frac{\partial N^{ml}}{\partial x}+\ddot{y}\frac{\partial N^{ml}}{\partial y}\right)q_{e} \end{array}\right)$$

Equivalent distributed force is specified according to the Delta-Dirac function as [32]:(35)$$\begin{array}{l} p(x,y,t)=Q(t)\delta (x-\xi )\delta (y-\eta )-mN^{ml}\delta (x-\xi )\delta (y-\eta ){\ddot{q}}_{e}+2m\left(\dot{x}\frac{\partial N^{ml}}{\partial x}+\dot{y}\frac{\partial N^{ml}}{\partial y}\right)\delta (x-\xi )\delta (y-\eta ){\dot{q}}_{e}+\ldots \\ +m\left({\dot{x}}^{2}\frac{\partial^{2}N^{ml}}{\partial x^{2}}+{\dot{y}}^{2}\frac{\partial^{2}N^{ml}}{\partial y^{2}}+2\dot{x}\dot{y}\frac{\partial^{2}N^{ml}}{\partial x\partial y}+\ddot{x}\frac{\partial N^{ml}}{\partial x}+\ddot{y}\frac{\partial N^{ml}}{\partial y}\right)\delta (x-\xi )\delta (y-\eta )q_{e} \end{array}$$

The nodal force vector of the element is computed from the distributed force *p*(*x*,*y*,*t*) applied on the element as following:(36)$$\begin{array}{l} F_{e}(t)=\int_{A_{t}}{\left(N^{ml}\right)}^{T}p(x,y,t)dA_{t}=\int_{A_{t}}\delta (x-\xi )\delta (y-\eta ){\left(N^{ml}\right)}^{T}Q(t)dA_{t}-\ldots \\ -m\int_{A}\delta (x-\xi )\delta (y-\eta ){\left(N^{ml}\right)}^{T}N^{ml}{\ddot{q}}_{e}dA_{t}\\ -2m\int_{A_{t}}\delta (x-\xi )\delta (y-\eta )\left(\dot{x}\frac{\partial N^{ml}}{\partial x}+\dot{y}\frac{\partial N^{ml}}{\partial y}\right){\left(N^{ml}\right)}^{T}{\dot{q}}_{e}dA_{t}\\ -m\int_{A_{t}}\delta (x-\xi )\delta (y-\eta )\left({\dot{x}}^{2}\frac{\partial^{2}N^{ml}}{\partial x^{2}}+{\dot{y}}^{2}\frac{\partial^{2}N^{ml}}{\partial y^{2}}+2\dot{x}\dot{y}\frac{\partial^{2}N^{ml}}{\partial x\partial y}+\ddot{x}\frac{\partial N^{ml}}{\partial x}+\ddot{y}\frac{\partial N^{ml}}{\partial y}\right)N^{T}{\dot{q}}_{e}dA_{t} \end{array}$$
or:(37)$$F_{e}(t)=P_{e}(t)-M_{e}^{ml}{\ddot{q}}_{e}-C_{e}^{ml}{\dot{q}}_{e}-K_{e}^{ml}q_{e}$$
where:(38)$$P_{e}(t)=\int_{A_{t}}\delta (x-\xi )\delta (y-\eta ){\left(N^{ml}\right)}^{T}Q(t)dA_{t}$$
(39)$$M_{e}^{ml}=m\int_{A}\delta (x-\xi )\delta (y-\eta ){\left(N^{ml}\right)}^{T}N^{ml}{\ddot{q}}_{e}dA_{t}$$
(40)$$C_{e}^{ml}=2m\int_{A_{t}}\delta (x-\xi )\delta (y-\eta )\left(\dot{x}\frac{\partial N^{ml}}{\partial x}+\dot{y}\frac{\partial N^{ml}}{\partial y}\right){\left(N^{ml}\right)}^{T}{\dot{q}}_{e}dA_{t}$$
(41)$$K_{e}^{ml}=m\int_{A_{t}}\delta (x-\xi )\delta (y-\eta )\left({\dot{x}}^{2}\frac{\partial^{2}N^{ml}}{\partial x^{2}}+{\dot{y}}^{2}\frac{\partial^{2}N^{ml}}{\partial y^{2}}+2\dot{x}\dot{y}\frac{\partial^{2}N^{ml}}{\partial x\partial y}+\ddot{x}\frac{\partial N^{ml}}{\partial x}+\ddot{y}\frac{\partial N^{ml}}{\partial y}\right){\left(N^{ml}\right)}^{T}{\dot{q}}_{e}dA_{t}$$
with $N^{ml}$ described in Appendix D.

### 3.3. Differential Equation of a Triple-Layer Plate under a Moving Load

By substituting Equation (37) into Equation (31), we obtained the dynamic equation of the plate element as follows:(42)$$\left(M_{e}^{p}+M_{e}^{ml}\right){\ddot{q}}_{e}+\left(C_{e}^{p}+C_{e}^{ml}\right){\dot{q}}_{e}+\left(K_{e}^{p}+K_{e}^{ml}\right)q_{e}=P_{e}(t)$$

They are linear differential equations, which have the coefficient depending on time. In order to solve these equations, we used the Newmark-beta method [32].

## 4. Numerical Results

### 4.1. Accuracy Study

Example 1: in this example, the accuracy and efficiency of this approach for multi-layer plates are confirmed by comparison with the exact three-dimensional elasticity solution. The composite plate (0°–90°–0°) was subjected to the sinusoidal loading *P* = *P*_0_sin(π*x*/*a*)sin(π*y*/*b*), the material properties are *E*_1_ = 25*E*_2_, *G*_12_ = *G*_13_ = 0.5*E*_2_, *G*_23_ = 0.2*E*_2_, *ν*_12_ = *ν*_13_ = *ν*_23_ = 0.25. The non-dimensional displacement was calculated as ${\bar{w}}_{c}=\frac{100h^{3}E_{2}}{P_{0}a^{4}}w(\frac{a}{2};\frac{b}{2};0)$. The numerical results with different meshes were compared with analytical and semi-analytical solutions, as shown in Table 1. From these results, we have shown a verification of the proposed method with six different meshes (from 10 × 10 to 20 × 20 meshes).

Example 2: consider a 4-layer composite plate (0°–90°–0°–90°) with the thickness of *h* (the thickness of each layer is *h*/4). The material properties and moving load parameters are the same as in Example 1. The plate was fully simply supported. The authors used the calculation program by considering the top layer and the bottom layer as two composite layers with fiber angles of 0° and 90°, respectively; the core layer becomes the 2-layer composite plate (90°–0°); and the shear coefficient of the shear connectors was large enough. Then, the 3-layer composite plate with shear connectors became a 4-layer composite plate. The non-dimensional displacement and the non-dimensional stress (of the center point of the structure) were computed using the following formulas: ${\bar{w}}_{c}=\frac{100h^{3}E_{2}}{P_{0}b^{3}}w(\frac{a}{2};\frac{b}{2};0);\;{\bar{\sigma }}_{x}=\frac{h^{2}}{P_{0}b^{2}}\sigma_{x}(\frac{a}{2};\frac{b}{2};0)$. The numerical results compared with [36] (based on analytical solution of layer wise theory), are presented in Table 2. From this example, we can see that the result of this work is in agreement with the results of the layer wise theory.

Example 3: consider a three-layer (0°–90°–0°) composite plate with a length of *a* = *b*; a thickness of *h* = *a*/20; a modulus of elasticity of *E*_1_ = 144.84 GPa, *E*_2_ = 9.65 GPa, *G*_12_ = *G*_13_ = 4.136 GPa, *G*_23_ = 3.447 GPa; a Poisson’s ratio of *ν*_12_ = *ν*_13_ = *ν*_23_ = 0.25; and a mass density of *ρ* = 1389.297 kg/m^3^. The plate was subjected to a moving concentrated force of *P*_0_ = 100 N with a constant speed of *v*_0_ = 40 m/s along the line segment *y* = *b*/2, as in reference [34]. The plate was simply supported at all edges. We verified our established program on the triple-layer plate, each layer had the thickness of 1/3 plate thickness and the shear coefficient of the shear connectors was very high. In this case, the triple-layer plate was considered as a three-layer composite, the result of this method (using the mesh 16 × 16) was compared with reference [34], as in Figure 2.

The non-dimensional center deflection of the plate is defined as:(43)$$W_{C}=\frac{E_{1}h^{3}}{P_{0}a^{2}}w(a/2,b/2,0),\;T=t/t_{f},t_{f}=a/v_{0}$$

Example 4: consider a simple supported rectangular plate with the pinned-free-pinned-free (SFSF—two short edges are pinned and two long edges are free) boundary condition with a length of *a =* 1 m; a width of *b = a*/2; a thickness of *h = a*/100; a modulus of Elasticity of *E =* 206.8 GPa; a Poisson’s ratio of *ν =* 0.29; a mass density of *ρ* = 7820 kg/m^3^ under a moving load with a mass of *M* = 2.3 kg along *y = b*/2, as in reference [33]. We then verified our established program on the triple-layer plate with three layers having the same material parameters. Each layer had the thickness of 1/3 plate thickness and the shear coefficient of the shear connector was very large. In this case, the triple-layer plate was considered as isotropic, and the result was compared with reference [33], as in Figure 3.

From Figure 2 and Figure 3, we can recognize that deflections of center points in our work are similar to references [33] and [34] in both shape and magnitude. This proves the reliability of our program.

### 4.2. Numerical Results of the Dynamic Analysis of Triple-Layer Composite Plates with Shear Connectors

Example 3: consider a triple-layer composite rectangular plate with following parameters: fixed length *a* = 10 m; width *b*, thickness *h* = *a*/50; thickness of the middle layer *h_s_*; thickness of the other two layers *h_a_* = *h_c_*; modulus of elasticity of the three layers *E_c_* = 8 GPa, *E_t_* = *E_b_* = 12 GPa; Poisson’s ratios *ν_c_* = 0.33; *ν_t_* = *ν_b_* = 0.2; and mass densities of three layers *ρ_c_* = 700 kg/m^3^; *ρ_t_* = *ρ_b_* = 2300 kg/m^3^; shear coefficients *k_tc_* = *k_cb_* = *k_s_* of the shear connector under moving load of mass *M* = 2.3 kg. Then, the non-dimensional deflections along *z* direction of the plate center point are:(44)$$\begin{array}{l} \bar{w}=\frac{10{h_{0}}^{3}E_{c}}{Mga^{2}(1-\nu_{c}^{2})}w\left(\frac{a}{2},\frac{b}{2},0\right);{\bar{u}}_{c}=\frac{10{h_{0}}^{3}E_{c}}{Mga^{2}(1-\nu_{c}^{2})}u_{c}\left(\frac{a}{2},\frac{b}{2},-\frac{h_{c}}{2}\right);{\bar{v}}_{c}=\frac{10{h_{0}}^{3}E_{c}}{Mga^{2}(1-\nu_{c}^{2})}v_{c}\left(\frac{a}{2},\frac{b}{2},-\frac{h_{c}}{2}\right);\\ \;\bar{v}=\frac{T{h_{0}}^{3}E_{c}}{Mga^{2}(1-\nu_{c}^{2})}v\left(\frac{a}{2},\frac{b}{2},0\right);\;{\bar{\sigma }}_{x}=\frac{10^{3}{h_{0}}^{4}}{Mga^{2}}\sigma_{x}\left(\frac{a}{2},\frac{b}{2},z\right)\;;\;{\bar{\sigma }}_{xy}=\frac{10^{3}{h_{0}}^{4}}{Mga^{2}}\sigma_{xy}\left(\frac{a}{2},\frac{b}{2},z\right);\; \end{array}$$
with *h*_0_ = *a*/50 (m) and *T* = 2 (s).

#### 4.2.1. Influence of the Modulus of Rigidity of Shear Connectors

We tested a fully simply supported (SSSS) square plate (*b = a*). The moving load moved along *y = b*/2 with velocity *v =* 5 m/s. The thicknesses of the three layers was *h_c_ = h*/2*; h_t_ = h_b_ = h*/4. Shear moduli of the shear connector were *k_s_* = 10^2^, 10^4^, 10^6^, 10^8^, 10^10^, 10^12^, 10^14^, 10^16^ Pa. Non-dimensional deflection velocity and stress of the plate center point are shown in Figure 4, maximum deflections and velocities of the plate center point are illustrated in Table 3. Nondimensional stress of the plate center point when the load moves to there is shown in Figure 5.

From Figure 4 and Table 3, we can conclude that when the shear modulus of the shear connector is increased from 10^2^ to 10^14^ Pa, the deflection and velocity of the plate center point, and the stress of the center point of the structure as a function of thickness in z-direction, decreased and decreased significantly at *k_s_* = 10^10^ Pa (above 10^10^ Pa, the deflection was nearly constant). With *k_s_* in the interval (10^2^–10^6^), the deflection reached its maximum. We can easily see the stress jumping at the contact surfaces in Figure 5. When k_s_ was small, the stress jumping was high. Therefore, depending on different cases in practice, we can choose correspondingly the suitable shear modulus for reducing the vibration of the plate.

#### 4.2.2. Influence of the Ratio *h_c_*/*h_t_*

In the following experiment, we tested a fully simply supported (SSSS) square plate (*b* = *a*). The shear modulus of the shear connector was *k_s_* = 50 MPa. The moving load travelled along *y* = *b*/2, with velocity *v* = 5 m/s, *h_c_/h_t_* = 1, 2, 4, 6, 8, 10, 12, 14 considered (the plate thickness *h* = *h_b_* + *h_c_* + *h_t_* was fixed). Non-dimensional deflection and the velocity of the plate center point are shown in Figure 6. Maximum deflections and velocities of the plate center point are illustrated in Table 4.

From Figure 6 and Table 4, we can see that when the ratio *h_c_/h_t_* was increased from 8 to 14 (i.e., increasing the thickness of the middle layer since the plate thickness *h* is fixed), deflection and velocity of the plate center point decreased significantly. This proved that the middle layer, which had smaller elastic modulus than the other two layers, could absorb vibration better and thus reduce deflection and velocity of the plate center point as a function of thickness in *z*-direction. Therefore, we proposed that we do not have to use material with large modulus of elasticity to decrease the vibration. Instead, we can use a triple-layer plate (with a shear connector) in which the middle layer has a smaller elastic modulus with a specific thickness ratio. Here, we obtain an interesting property: the vertical displacement *w* decreased when the ratio *h_c_*/*h_t_* increased, and in contrast, the horizontal displacements *u_c_* and *v_c_* (at the contact surfaces) increased. This can be explained by the strain energy focusing on the membrane direction.

#### 4.2.3. Influence of the Ratio *a*/*h*

Let us consider a fully simply supported (SSSS) square plate (*b* = *a*) with *h_c_* = *h*/2*, h_t_* = *h_b_* = *h*/4. Shear modulus of the shear connector was *k_s_* = 50 MPa. The moving load travelled along *y* = *b*/2 with velocity *v* = 5 m/s. *a/h* = 30, 40, 50, 60, 70, 80, 90, 100 used. Non-dimensional deflection and the velocity of the plate center point are shown in Figure 7, maximum deflections and velocities of the plate center point are illustrated are Table 5.

From Figure 7 and Table 5, we can recognize that when the plate thickness was increased from *a*/100 to *a*/30, deflection, velocity and stress of the plate center point decreased and decreased significantly in the range of *a*/50 to *a*/30.

#### 4.2.4. Influence of the Moving Load Velocity

Let us consider a fully simply supported (SSSS) square plate (*b* = *a*) with *h_c_* = *h*/2*, h_t_* = *h_b_* = *h*/4. Shear modulus of the shear connector was *k_s_* = 50 MPa. The moving load travelled along *y* = *b*/2 with velocities *v* = 5, 10, 15, 20, 25, 30, 35, 40 m/s. Non-dimensional deflection and the velocity of the plate center point are shown in Figure 8, maximum deflections and velocities of the plate center point are illustrated in Table 6.

From Figure 8 and Table 6, we can find that when the velocity of the moving load was increased from 5 to 40 m/s, deflection, velocity of the plate center point as a function of thickness in *z*-direction increased and slightly increased in the range of 5–15 m/s and 20–40 m/s.

#### 4.2.5. Influence of Mass Density of the Core Layer

Let us consider a four edged simply supported (SSSS) square plate (*b* = *a*) with *h_c_* = *h*/2*, h_t_* = *h_b_* = *h*/4. Shear modulus of the shear connector was *k_s_* = 50 MPa. The moving load travelled along *y* = *b*/2 with velocity *v* = 5 m/s. Mass densities of three layers *ρ_t_* = *ρ_b_* = 2300 kg/m^3^ and *ρ_c_* = 700, 1000, 1500, 2000, 2300 kg/m^3^ were used. Non-dimensional deflection, velocity and stress of the plate center point are shown in Figure 9, maximum deflections and velocities of the plate center point are illustrated in Table 7.

From Figure 9 and Table 7, we can find that when the mass density of the core-layer was increased from 700 to 2300 kg, deflection and velocities of the plate center point as a function of thickness in *z*-direction were almost not changed. Therefore, in order to reduce the mass of the plate we can use a triple-layer plate with shear connectors, where the core layer has a smaller mass density than other layers.

#### 4.2.6. Influence of Modulus of Elasticity

Let us consider a fully simply supported (SSSS) square plate (*b* = *a*) with *h_c_* = *h*/2, *h_t_* = *h_b_* = *h*/4. The shear modulus of the shear connector was *k_s_* = 50 MPa. The moving load travelled along *y* = *b*/2 with velocity *v* = 5m/s. A modulus of elasticity of three layers *E_t_* = *E_b_* = 12 GPa and *E_c_* = 8, 9, 10, 12 GPa, was used. Non-dimensional deflection and velocity of the plate center point is shown in Figure 10, maximum deflections and velocities of the plate center point are illustrated in Table 8.

From Figure 10 and Table 8, we can find that when modulus of elasticity core-layer was increased from 8 to 12 GPa, deflection, velocity of the plate center point and the stress of center point as a function of thickness in z-direction were slightly decreased.

## 5. Conclusions

By using published theories of multi-layered beams, Mindlin’s plate theory and finite element modelling, we simulated the forced vibration of a triple-layer plate with layers connected by shear connectors subjected to a moving load. The influences of some structural parameters on the dynamic response of the plate were also examined in the paper. We found that, in the regarded plate, the shear coefficient of the shear connector played a crucial role. Specifically, when the shear coefficient was sufficiently large, the plate could be considered a sandwich plate. The flexibility of the shear coefficient helped engineer a plate with the desired mechanical properties. From the results of numerical tests, we proposed that to reduce plate vibration, the elastic modulus of the middle layer should be smaller than the outside two layers, and the thickness of the middle layers should be 20–30 times larger than the two outside layers.

## Figures and Tables

**Figure 1 materials-12-00598-f001:**
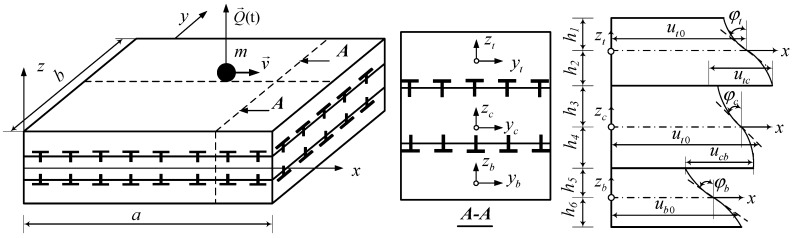
Model of a triple-layer composite plate with a shear connector subjected to a moving load.

**Figure 2 materials-12-00598-f002:**
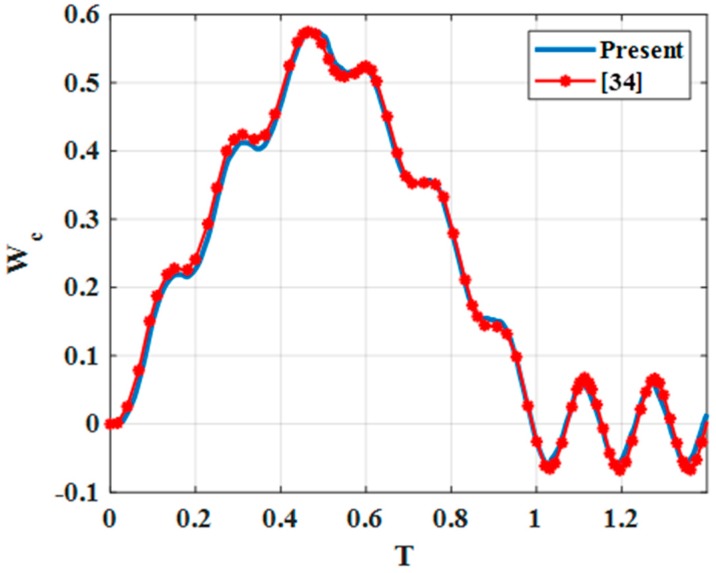
Non-dimensional center deflection of the plate under a moving concentrated load.

**Figure 3 materials-12-00598-f003:**
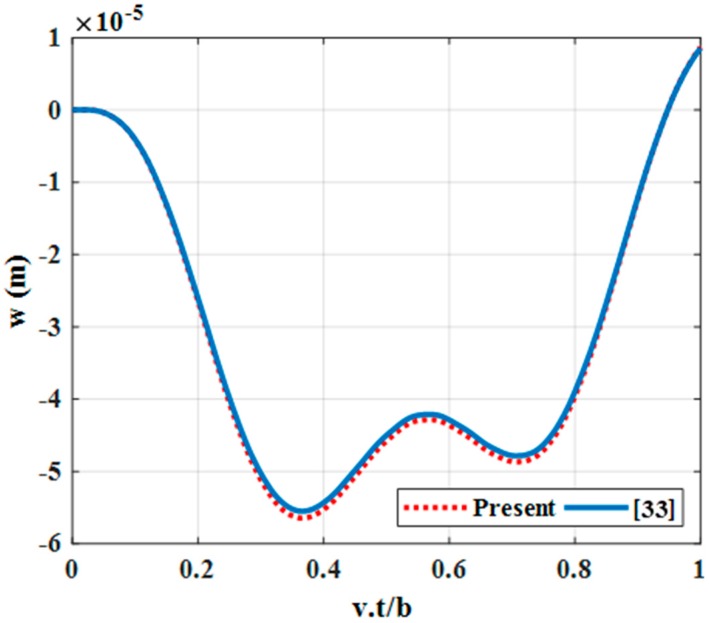
Dynamic deflections of the center point of the SFSF rectangular plate versus time.

**Figure 4 materials-12-00598-f004:**
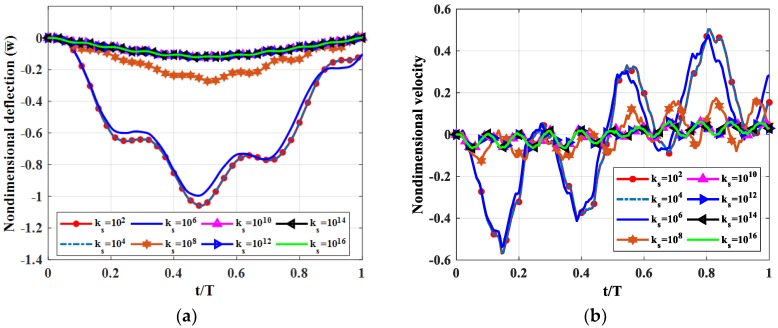
Dynamic deflections of the center point of the plate versus time for different shear moduli of the shear connector. (**a**): Nondimensional deflection $\bar{w}$ versus time, (**b**): Nondimensional velocity $\bar{v}$ versus time.

**Figure 5 materials-12-00598-f005:**
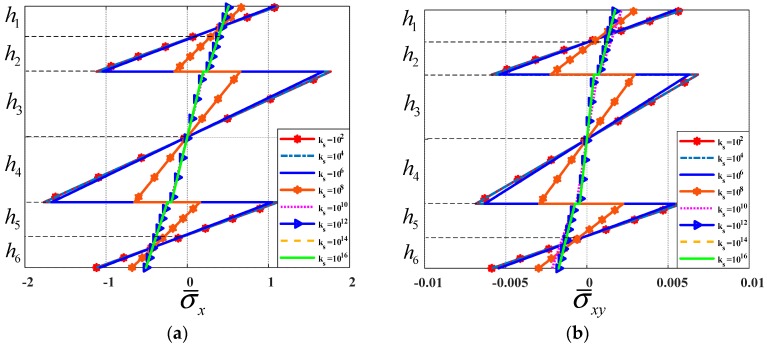
Non-dimensional stress of the plate center point when the load moves to the plate center point. (**a**): Nondimensional stress ${\bar{\sigma }}_{x}$ versus thickness, (**b**): Nondimensional stress ${\bar{\sigma }}_{xy}$ versus thickness.

**Figure 6 materials-12-00598-f006:**
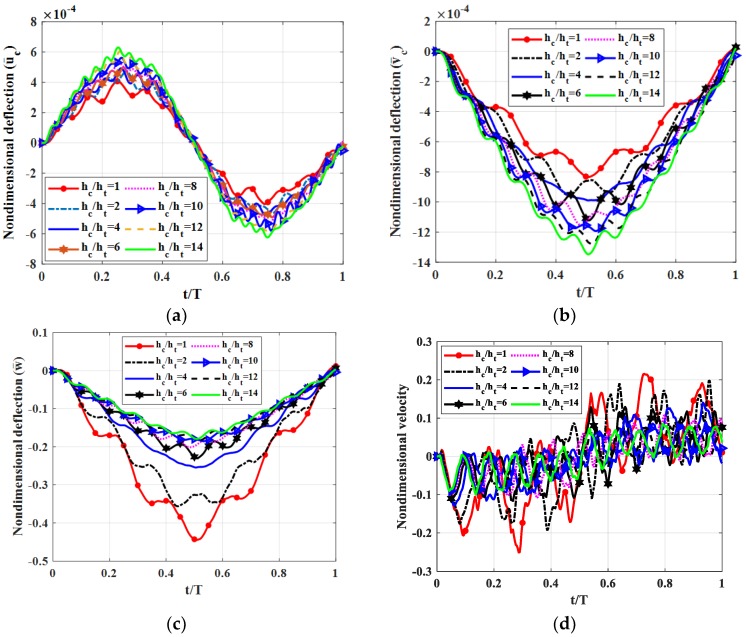
Dynamic deflections of center point of the plate versus time for different ratios *h_c_*/*h_t_*. (**a**): Nondimensional deflection ${\bar{u}}_{c}$ versus time, (**b**): Nondimensional deflection ${\bar{v}}_{c}$ versus time, (**c**): Nondimensional velocity $\bar{w}$ versus time, (**d**): Nondimensional velocity $\bar{v}$ versus time.

**Figure 7 materials-12-00598-f007:**
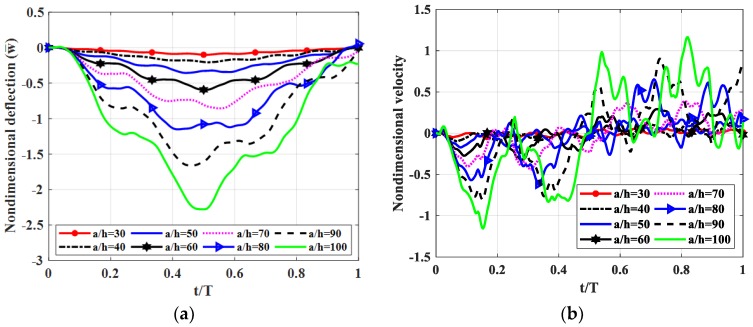
Dynamic deflections of center point of the plate versus time for different ratios *a/h.* (**a**): Nondimensional deflection $\bar{w}$ versus time, (**b**): Nondimensional velocity $\bar{v}$ versus time.

**Figure 8 materials-12-00598-f008:**
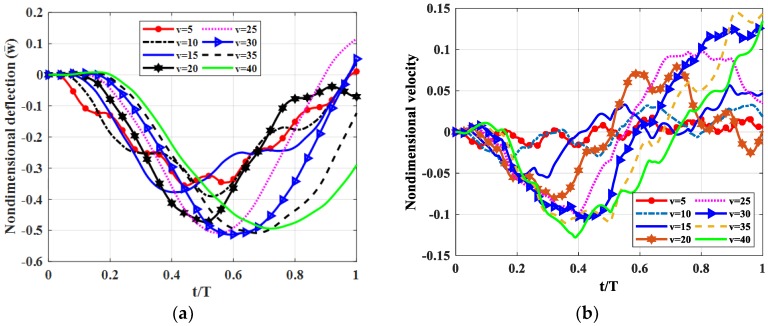
Dynamic deflections of the center point of the plate versus time for different velocities. (**a**): Nondimensional deflection $\bar{w}$ versus time, (**b**): Nondimensional velocity $\bar{v}$ versus time.

**Figure 9 materials-12-00598-f009:**
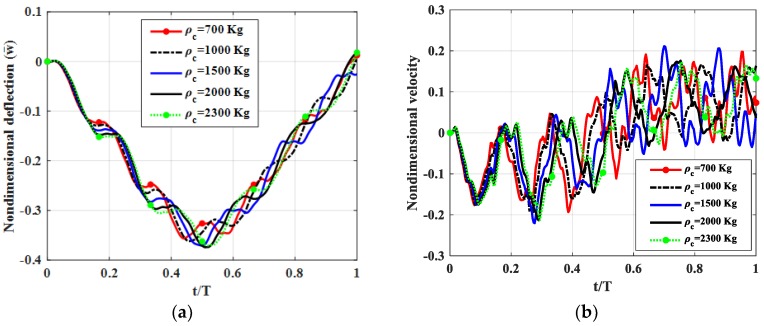
Dynamic deflections of the center point of the plate versus time for different *ρ_c_.* (**a**): Nondimensional deflection $\bar{w}$ versus time, (**b**): Nondimensional velocity $\bar{v}$ versus time.

**Figure 10 materials-12-00598-f010:**
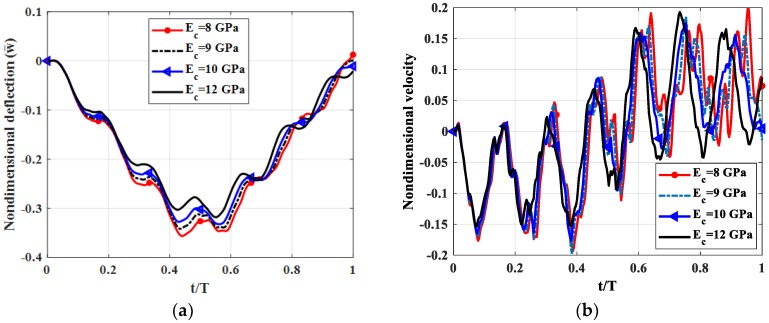
Dynamic deflections of the center point of the plate versus time for different *E_c_.* (**a**): Nondimensional deflection $\bar{w}$ versus time, (**b**): Nondimensional velocity $\bar{v}$ versus time.

**Table 1 materials-12-00598-t001:** Maximum deflections ${\bar{w}}_{c}$ versus mesh density (*a*/*h* = 10).

Mesh	10 × 10	12 × 12	14 × 14	16 × 16	18 × 18	20 × 20
Present	0.7250	0.7312	0.7349	0.7528	0.7529	0.7529
[34]			0.7530			
[35]			0.7530			

**Table 2 materials-12-00598-t002:** Maximum deflection ${\bar{w}}_{c}$ and stress ${\bar{\sigma }}_{x}$ versus length-to-width ratio (*a*/*h* = 10).

*a*/*b*		1	2	3	4	6
${\bar{w}}_{c}$	Present	−0.7582	−1.4387	−1.5684	−1.6032	−1.6170
	[36]	−0.7541	−1.4451	−1.5720	−1.6060	−1.6256
${\bar{\sigma }}_{x}$	Present	0.5225	0.3178	0.1734	0.1100	0.0621
	[36]	0.5211	0.3166	0.1704	0.1078	0.0600

**Table 3 materials-12-00598-t003:** Maximum deflections and velocities of the plate center point versus time.

Maximum Values	*k_s_* = 10^2^	*k_s_* = 10^4^	*k_s_* = 10^6^	*k_s_* = 10^8^	*k_s_* = 10^10^	*k_s_* = 10^12^	*k_s_* = 10^14^	*k_s_* = 10^16^
${\bar{w}}_{\max}$	1.0581	1.0574	0.9951	0.2732	0.1240	0.1215	0.1215	0.1215
${\bar{v}}_{\max}$	0.5680	0.5677	0.5380	0.1278	0.0656	0.0647	0.0647	0.0647

**Table 4 materials-12-00598-t004:** Maximum deflections and velocities of the plate center point of the plate versus time for different ratios *h_c_*/*h_t_*.

Maximum Values	*h_c_*/*h_t_* = 1	*h_c_*/*h_t_* = 2	*h_c_*/*h_t_* = 4	*h_c_*/*h_t_* = 6	*h_c_*/*h_t_* = 8	*h_c_*/*h_t_* = 10	*h_c_*/*h_t_* = 12	*h_c_*/*h_t_* = 14
${\bar{w}}_{\max}$	0.4449	0.3564	0.2545	0.2302	0.2014	0.1855	0.1826	0.1796
${\bar{u}}_{c}^{\max}\times 10^{-4}$	4.0349	4.6229	4.8121	4.9093	5.2788	5.7961	5.7851	6.2406
${\bar{v}}_{c}^{\max}\times 10^{-4}$	8.3483	9.5245	9.8925	11.23073	11.5314	11.9324	12.7891	13.4772
${\bar{v}}_{\max}$	0.2509	0.1934	0.1283	0.1162	0.1087	0.0985	0.1020	0.0998

**Table 5 materials-12-00598-t005:** Maximum deflections, velocities and stress of the plate center point versus time for different ratios *a/h.*

Maximum Values	*a/h* = 30	*a/h* = 40	*a/h* = 50	*a/h* = 60	*a/h* = 70	*a/h* = 80	*a/h* = 90	*a/h* = 100
${\bar{w}}_{\max}$	0.1015	0.2069	0.3564	0.5964	0.8566	1.1515	1.6623	2.2777
${\bar{v}}_{\max}$	0.0517	0.0977	0.1934	0.3280	0.4420	0.6170	0.8025	1.1552

**Table 6 materials-12-00598-t006:** Maximum deflections, velocities and stress of the plate center point versus time for different velocities.

Maximum Values	*v* = 5	*v* = 10	*v* = 15	*v* = 20	*v* = 25	*v* = 30	*v* = 35	*v* = 40
${\bar{w}}_{\max}$	0.3564	0.3904	0.3772	0.4734	0.5083	0.5132	0.5081	0.4940
${\bar{v}}_{\max}$	0.0175	0.0409	0.0547	0.0801	0.1000	0.1046	0.1105	0.1282

**Table 7 materials-12-00598-t007:** Maximum deflections, velocities and stress of the plate center point versus time for different *ρ_c_*.

Maximum Values	*ρ_c_* = 700 kg/m^3^	*ρ_c_* = 1000 kg/m^3^	*ρ_c_* = 1500 kg/m^3^	*ρ_c_* = 2000 kg/m^3^	*ρ_c_* = 2300 kg/m^3^
${\bar{w}}_{\max}$	0.3564	0.3619	0.3711	0.3742	0.3707
${\bar{v}}_{\max}$	0.1934	0.1950	0.2202	0.2131	0.2136

**Table 8 materials-12-00598-t008:** Maximum deflections, velocities and stress of the plate center point versus time for different *E_c_.*

Maximum Values	*E_c_* = 8 GPa	*E_c_* = 9 GPa	*E_c_* = 10 GPa	*E_c_* = 12 GPa
${\bar{w}}_{\max}$	0.3564	0.3418	0.3418	0.3181
${\bar{v}}_{\max}$	0.1934	0.1974	0.1763	0.1542

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
