# Peer review of "A Finite Element Model for Dynamic Analysis of Triple-Layer Composite Plates with Layers Connected by Shear Connectors Subjected to Moving Load"

_materials, 2019, doi:10.3390/ma12040598_

Round 1
Reviewer 1 Report
- Eq. 1 is it not a limitation to consider w constant for all layers? In-plane displacements are considered different layer by layer but w not. Please comment.
- line 121: shell element? is not a plate element studied?
- line 150 "stud"? Please correct typo
- line 150 what is the meaning of "very large shear coefficient"? Please specify.
- line 155 since the plate is not squared please specify better (SFSF) which edge is supported and which one is free
- line 170 please avoid capital K for kg.
- Comparison are provided with [24] but first of all convergence analysis of the present dynamic and FE approach should be reported.
- Moreover, a comparison with another composite ref should be provided because [24] considers isotropic materials and it is not very convincing considering "w" as constant though all the layers.
Please check reference papers regarding laminated composite structures with layer-wise formulations.
Author Response
Thank you for your comments, Please find in the attachment our point-by-point response

Reviewer 2 Report
The paper presents an original numerical procedure that allows to model a triple-layered composite plates with accounting for shear connectors and moving load. Proposed method may be an effective engineering tool for designing composite plates involving the shear connectors. In order to prove a reliability of the method described in the paper the Authors compared their results with results presented in literature obtained by using different approach. The comparison was made for the homogenous plate by prescribing the same material properties for the core and outside layers, in this case the results are in good agreement. Then an influence of several structural parameters on the dynamic response of the plate was investigated. Here, the presented examples and obtained results need further clarification. My comments and questions are listed below.
The most important remarks:
1. Section 4.2.1 - table 1 suggests that the deflection is increasing with increasing stiffness of the shear connector – it is opposite to the data presented in the figure 4 and the comment starting form line 182. Moreover the figure 4 shows that maximum displacement value is approximately 1 but the table 1 states that the maximum displacement is 2.2777. Mentioned inconsistencies must be verified and corrected (I suppose that there is something wrong with the Table 1).
2. Section 4.2.2 - The Authors provide conclusion that deflection and velocity decrease significantly with the increasing thickness of the middle layer. In the case of Young moduli of the layers and the fixed total thickness (please confirm if the total thickness is fixed or rather the thickness of the outside layer is fixed?), which were assumed, this result is quite surprising. Therefore in order to prove reliability of the obtained results and formulate a general conclusion I recommend to test additionally an influence of different ratios of elastic moduli and mass densities corresponding to the different layers (especially due to fact that considered contrast between the Ec and Et=Eb is relatively small).
3. Section 4.2 – what is the purpose of using the nondimensional quantities and how the equation 43 is established ? Moreover the indices connected with modulus of elasticity (Es, Ea=Ec) and mass densities of different layers are denoted differently than in the previous sections where index c was corresponding to the middle layer and t and b to the top and bottom layers respectively.
4. Equation 42 corresponds to the single finite element, I have not noticed any other equation corresponding to the system of finite elements. Therefore my question is if computations were performed on a basis of just one finite element or rather an assembly of several finite elements? How many elements have been used ? Have the Authors investigated the finite element mesh density influence on the results accuracy?
5. Line 146 - I suppose that given thickness value is incorrect here – in the case when the overall dimensions of plate are 10x10 m and load magnitude is 5.1 kN the thickness 0.00655 m will may lead to very large displacements. Maybe a correct value of thickness is 0.0655 m ? – compare with [24].
6. Analysis of stresses in the plate and shear connector has not been presented at all. In my opinion a discussion on the stresses in different layers and shear connector will emphasize the usefulness of proposed approach and make the manuscript more interesting for readers.
7. Line 237- I suggest to change the sentence “experimental results” for example to “results of numerical tests”. The paper presents only numerical results therefore the word “experimental” may confuse the readers.
Minor points:
1. Keywords - I suggest to change the keyword “blast load” to “moving load” which is more suitable for this paper.
2. The introduction section may be enriched by referring to the studies connected with three-dimensional finite element analysis of plates with shear connector like for example:
Jia-Bao Yan, Wei Zhang. Numerical analysis on steel-concrete-steel sandwich plates by damage plasticity model: From materials to structures, Construction and Building Materials, 149:801-815
3. Line 33, A word “ratio” should be added in the following way: “stiffness-to-weight ratio”
4. Line 78 - ”..local coordinates oxyt, oxyc, oxy” - here probably should be oxyt, oxyc, oxyb
5. Figure 4 – the curve representing ks=1010 is hardly noticeable - some marker should be added to make it clearly visible.
6. Line 185 – Statement “Therefore, we only need to manufacture….reduce the vibration of the plate” should be rewritten due to poor language style.
7. Line 189 – The unit of the modulus of the shear connector is missing.
8. Equation 1 – Sentence “(vói k=t,c,b)” should be rewritten in English
9. Equation 25 – Are a different values of k appropriate ? – I mean that once k=c,s,a and another time k=t,c,b.
Author Response

(The authors gave the same response as above.)

Reviewer 3 Report
This paper presents structural models to solve dynamics problems of composite structures. Although technically sound, this paper does not present enough novelties for publication. In fact,
1) The main novelty claimed by the authors is the inclusion of shear deformable interfaces between layers. The approach they use is an equivalent single layer one and has many examples in literature. Moreover, more advanced techniques are already available for the same problems, such as the layer-wise approach or higher-order models with transverse stretching.
2) the literature review lacks very important contributions, such as GBT, VABS, CUF.
3) The choice of the authors to restrict the formulation and results to 3 layers is very questionable given that much more layers are used in real applications.
4) The results do not present any stress distributions and no investigations concerning the effect of typical design parameters such as the stacking sequence.
Author Response

(The authors gave the same response as above.)

Round 2
Reviewer 1 Report
Comments provided by the authors improved the manuscript in my opinion and provide a sufficient justification to the present work, which should be accepted for publication in the present form.
Author Response
Thank you very much for your comment
Reviewer 2 Report
The authors have improved the manuscript significantly and have convincingly responded to all my comments. However I found some minor mistakes which should be corrected:
1. Figure 2 – inconsistency of the references - figure refers to the [28] while in the text there is the reference to [34]
2. Figure 3 – inconsistency of the references - figure refers to the [24] while in the text there is the reference to [33]
3. Line 231 – The sentence “When ks is small, the stress jumping is also small” should be rewritten - it is opposite to the results provided in the Fig. 5.
4. Line 232 – “…modulus of about 1010 are optimal for reducing…” – I suggest to avoid a word “optimal” in this case. In my opinion, it will be better to provide a simple comment that shear modulus of the shear connector equal 1010 is enough to provide a significant reduction of the vibration.
5. Line 160 P=P.sin…. (the dot should be removed)
6. Table 7 – densities should be expressed in kg/m3
7. Question 9 from the first review – Equation 25 - please verify if the indices c,s,a and t,c,b have been unified properly in the revised manuscript.
Author Response
Many thanks for your comment, our justification and explanations are as follows:
Question 1: Figure 2 – inconsistency of the references - figure refers to the [28] while in the text there is the reference to [34]
Answer: We are so sorry for this mistake. Figure 2 has been corrected.
Question 2: Figure 3 – inconsistency of the references - figure refers to the [24] while in the text there is the reference to [33]
Answer: Figure 3 was corrected by authors.
Question 3: Line 231 – The sentence “When ks is small, the stress jumping is also small” should be rewritten - it is opposite to the results provided in the Fig. 5.
Answer: Many thanks for reviewer suggest. We have been rewritten this sentence (Line 234).
Question 4: Line 232 – “…modulus of about 1010 are optimal for reducing…” – I suggest to avoid a word “optimal” in this case. In my opinion, it will be better to provide a simple comment that shear modulus of the shear connector equal 1010 is enough to provide a significant reduction of the vibration.
Answer: The suggestion of reviewer is exact. This sentence was replaced (Lines 234-236).
Question 5: Line 160 P=P.sin…. (the dot should be removed)
Answer: This dot was removed (Line 161).
Question 6: Table 7 – densities should be expressed in kg/m3
Answer: Thanks for your suggestion. Table 7 was corrected.
Question 7: Question 9 from the first review – Equation 25 - please verify if the indices c,s,a and t,c,b have been unified properly in the revised manuscript.
Answer: Equation 25 has been corrected.
We have revised our paper as reviewer’s suggestions. We also have rechecked our paper very carefully, and we have made several necessary changes as well as added some needed new references for the reader’s better understanding. All the changes have been highlighted in yellow color.
We have tried our best to show the most important findings in the manuscript and we hope that our effort will be appreciated.
Reviewer 3 Report
The revised version of the manuscript can be accepted
Author Response
Thank you very much for your comment